# Adverse Outcome Pathway ‘Footprinting’: A Novel Approach to the Integration of 21st Century Toxicology Information into Chemical Mixtures Risk Assessment

**DOI:** 10.3390/toxics11010037

**Published:** 2022-12-30

**Authors:** Jason C. Lambert

**Affiliations:** United States Environmental Protection Agency, Office of Research and Development, Center for Computational Toxicology and Exposure, Cincinnati, OH 45268, USA; lambert.jason@epa.gov; Tel.: +1-513-569-7078

**Keywords:** New Approach Methodologies (NAMs), Adverse Outcome Pathway (AOP), key event, chemical mixtures, human health risk assessment

## Abstract

For over a decade, New Approach Methodologies (NAMs) such as structure-activity/read-across, -omics technologies, and Adverse Outcome Pathway (AOP), have been considered within regulatory communities as alternative sources of chemical and biological information potentially relevant to human health risk assessment. Integration of NAMs into applications such as chemical mixtures risk assessment has been limited due to the lack of validation of qualitative and quantitative application to adverse health outcomes in vivo, and acceptance by risk assessors. However, leveraging existent hazard and dose–response information, including NAM-based data, for mixture component chemicals across one or more levels of biological organization using novel approaches such as AOP ‘footprinting’ proposed herein, may significantly advance mixtures risk assessment. AOP footprinting entails the systematic stepwise profiling and comparison of all known or suspected AOPs involved in a toxicological effect at the level of key event (KE). The goal is to identify key event(s) most proximal to an adverse outcome within each AOP suspected of contributing to a given health outcome at which similarity between mixture chemicals can be confidently determined. These key events are identified as the ‘footprint’ for a given AOP. This work presents the general concept, and a hypothetical example application, of AOP footprinting as a key methodology for the integration of NAM data into mixtures risk assessment.

## 1. Introduction

Human health risk assessment of environmental mixtures is inherently complex as there is often a lack of hazard and dose–response information for whole mixtures of concern, or for individual component chemicals. Further, while the scientific community has posited considerations for advancing mixture risk assessment such as evaluation of combined exposures and effects associated with ‘real-life’ chemical mixtures (e.g., singlet organics/inorganics, metals, polymers, UVCBs, etc.) encountered in environmental exposure sources, and integration of intermediate measures to account for mixture effects (e.g., mixture assessment factor), strategies and frameworks resulting in practical application are limited to date [1]. Part of the challenge is recapitulating what constitutes a representative or generalizable whole mixture exposure (A “whole mixture” entails the complete profile of parent chemicals, precursors, degredation products, and/or metabolites at individual constituent proportions consistent with that found in either environmental exposure sources, or experimentally designed in a laboratory.); this is due primarily to the complicated milieu of chemicals commonly present in environmental media (e.g., water, soil, air). Further, once exposed, internalized chemicals may transit, distribute, undergo metabolism and bioaccumulate within tissues in ways that lead to yet different ‘biological mixtures’ at a target site. Characterizing such environmental fate, toxicokinetic and toxicodynamic properties of mixtures of chemicals using traditional animal bioassay approaches is methodologically difficult and resource and time intensive.

New Approach Methodology (NAM) platforms and data, such as cell-based bioactivity assays/-omics (e.g., transcriptomics, proteomics, metabolomics) and read-across have been proposed for expediting human health risk assessment for over a decade [2,3,4,5,6]. However, formal NAM-based human health risk assessment application in the U.S. EPA, Health Canada, and European regulatory entities have relied primarily on read-across applications to inform evaluation of data-poor chemicals (for example, see Appendix A at: https://cfpub.epa.gov/ncea/pprtv/recordisplay.cfm?deid=339035, accessed on 15 December 2022; also see: https://www.canada.ca/en/health-canada/services/chemical-substances/fact-sheets/analogues-read-across-risk-assessment.html, accessed on 15 December 2022; the European Chemicals Agency provides significant resources supporting use of read-across at: https://echa.europa.eu/support/registration/how-to-avoid-unnecessary-testing-on-animals/grouping-of-substances-and-read-across; accessed on 16 December 2022). Compared to traditional long(er)-term bioassays of large numbers of animals, NAMs present more opportunities to test chemicals in in vitro cell based, short(er)-term in vivo, or ex vivo tissue-based assays more rapidly, across greater concentration ranges, and in animal sparing laboratory environments. Even with these advantages, NAM-derived data have not been integrated to a significant degree in regulatory or assessment applications for the evaluation of environmental chemicals, especially mixtures. As such, informing potential human health outcomes associated with exposure to chemical mixtures continues to be primarily reliant on single chemical-by-chemical hazard and dose–response information and component-based mixtures approaches [7,8].

An important consideration in the application of NAM-derived information in mixtures assessment is that with the exception of pharmaceuticals, hormone therapies, and some pesticides, most environmental chemicals were not typically designed or intended to specifically interact with biological targets in humans or ecological species. In general, in contrast to many pharmacological agents with relatively specific molecular or cellular targets or bioactivities, environmental chemicals commonly induce a complex profile of biological perturbations. Exposure to an environmental chemical at a given dose level may result in activation and/or inhibition of a diverse mix of receptor-dependent and –independent pathways that lead to complex bioactivities and ultimately a broad landscape of health outcomes. The myriad of kinetic and dynamic pathway perturbations that orchestrate an adverse tissue or organ response have traditionally been evaluated for human health assessment under a number of related constructs in the U.S. EPA, such as mode or mechanism of action [8,9,10]. However, the toxic mode/mechanism of action concept has subsequently been integrated and expanded under the Adverse Outcome Pathway (AOP) paradigm [11]. AOPs were first developed for the purpose of characterizing toxicity pathways in ecological species [12] but have since been posited as the preferred modality for structured qualitative and quantitative description of biological events spanning the entire exposure to outcome continuum, from a molecular initiating event (MIE), through one or more intermediate key events culminating in an adverse health outcome at the organismal level, up to population level dynamics for humans and ecological species [13,14,15,16]. While many environmental chemicals induce complex biological perturbations at a mechanistically granular level of organization, AOPs are commonly represented as unbranched (i.e., linear) ‘if this then that’ flow of key events [11,12,13,14,15,16]. More recently, the focus in AOP research has shifted to AOP network mapping, where the interaction of two or more AOPs are causally linked to the elicitation of an adverse health outcome, including interactions between key events of contributing AOPs [17,18].

One of the initial steps in any component-based mixtures assessment is grouping chemicals based on their similarity of toxic action [8,9]. Chemicals exhibiting a toxicological commonality have been described previously as sharing a common adverse outcome [19], a common mode of action [8], and/or are characterized as a common mechanism group [10]. An important nuance in the various pathway-based constructs is whether toxicokinetics are or are not considered part of an adverse outcome continuum or mode or mechanism of toxic action. Indeed, Price et al. (2019) separated toxicokinetics and toxicodynamics associated with key-event based chemical behavior into separate categories in an integrated adverse exposure-outcome pathway (AEP-AOP) schema [20]. For the purposes of the AOP footprinting approach, toxicokinetics are considered part of a source-to-outcome continuum that can be leveraged in evaluation(s) of similarity across mixture component chemicals.

Individual chemical pathway perturbation data from NAM platforms may facilitate predictions or inferences of adverse health outcome(s) anticipated from a mixture exposure in the virtual absence of traditional apical effect toxicity data. Once chemicals are qualitatively grouped together, the greater challenge is determining whether and how the NAM-based concentration-response data can be used to inform mixtures dose–response assessment. For example, interpreting how a given level of perturbation in a non-apical toxicokinetic process or toxicodynamic effect translates to incidence and/or magnitude of a given adverse health outcome in an organism remains a challenge and a source of uncertainty when using such data in risk assessment.

Chemicals co-occurring in mixtures may interact chemically and/or biologically to produce effects that are greater than anticipated (e.g., synergy), less than expected (e.g., antagonism), or in an additive manner [8]. Additivity of chemicals in mixture may be dose or response additive. In the U.S. EPA’s Supplementary Guidance for Conducting Health Risk Assessment of Chemical Mixtures (2000), “chemicals can be considered as dose additive if each chemical can be thought of as a concentration or dilution of every other chemical in the mixture. The chemicals are assumed to behave similarly in terms of the primary physiologic processes (uptake, metabolism, distribution, elimination) as well as the toxicologic processes”. Conversely, response additivity is considered applicable when a biological response to one mixture chemical is seemingly unaffected by the presence of another; that is, they induce an adverse health outcome via pathways that are independent [8]. Importantly, additivity of mixture chemicals has historically been evaluated at the apical (phenotypic) end of an adverse outcome pathway (i.e., tissue/organ level effect). However, with growing attention on use of NAM-based data to inform risk assessment applications such as chemical mixture behavior, evaluation of additivity in biological systems has increasingly leveraged in vitro platforms (e.g., cell bioactivity; toxicogenomics). A recent review by Martin et al. systematically reviewed over 1200 existent mixture studies, spanning in vitro to in vivo study designs, available in the public domain, and found that a default assumption of dose (or concentration) addition is supported across a diverse landscape of chemical mixtures [21]. It should be cautioned however that evaluating hazard and dose–response further upstream from the apical adverse outcome may lead to more frequent interpretations of deviation from dose additivity (i.e., chemicals may not qualitatively look biologically similar simply because of the inherent diversity of cellular/molecular events at a mechanistic level of “hazard” understanding) [19]. Further, a given AOP may include modules or nodes of key events that appear to deviate from dose additivity at one or more point(s) along the pathway continuum yet exhibit dose additivity at other points along the continuum. Thus, a key to qualitative ‘hazard’ grouping of chemicals is the level of biological organization at which decisions regarding commonality are made. Likewise, a critical consideration for an AOP-based quantitative evaluation of chemical mixture additivity is where along the pathway, from MIE to adverse health outcome, are such determinations made.

While bioactivity pathways or networks are predominately diverse and complex at and/or immediately downstream of a MIE (e.g., kinases/phosphatases; G-protein coupled receptors [GPCRs]; oxidative stress/redox status), it has been empirically observed that there are critical molecular/cellular pathway junctions that serve as a convergence point for signaling and that these junctions are often a key event more proximally located to an adverse outcome. A prime example was illustrated in the NAS’ 2008 report on “Phthalates and Cumulative Risk Assessment: The Tasks Ahead” whereby several potential adverse reproductive/developmental health outcomes in males (primarily experimental rats) were proposed to occur via a number of different pathways (Figures 3 and 4 of the 2008 NAS report, [19]). While the mechanisms proceeding from the various MIEs (e.g., androgen receptor blockade or decreased androgen biosynthesis) entail unique bioactivities [22,23], they ultimately impinge on the same downstream pathway junction or key event, for example, decreased androgen signaling at the target tissue [19]. However, even with comprehensive qualitative characterization of the source-to-outcome continuum in AOPs, the greater challenge is how to quantify the biology across AOPs such that confident predictions of health outcomes can be made. Thus, qualitatively identifying AOPs and the critical intermediate key events and quantitatively characterizing the dose–response relationships, across different levels of biological organization, for potentially diverse landscapes of health outcomes associated with co-occurring environmental chemicals will be paramount to advancing mixtures risk assessment in an evolving toxicity data environment.

The AOP footprinting approach described below provides a novel framework for the integration of diverse data types and sources with existent U.S. EPA component-based mixtures risk assessment methodology (e.g., relative potency factor). Using qualitative information across potentially different levels of biological organization, and quantitative dose- or concentration-response data derived from NAM-based platforms, under an assumption of dose additivity, AOP footprinting may facilitate a fundamental evolution in chemical mixtures risk assessment.

## 2. Method: Adverse Outcome Pathway (AOP) Footprinting

The steps of the AOP footprinting approach involve multiple tasks or substeps and are described below. The general steps are: (1) Survey, assembly, and evidence mapping of existent toxicology data across mixture chemicals; (2) Assigning mixture chemicals into adverse health outcome groupings; (3) Comparative evaluation of toxicokinetic (TK) data between members of an adverse health outcome grouping (i.e., augmenting weight-of-evidence for data-poor mixture chemicals through comparison of TK data to anchor chemicals); (4) Identification and assembly of operant AOPs per health outcome; (5) AOP footprinting; and (6) Component-based mixture risk assessment (e.g., development of relative potency factors and associated AOP anchor chemical equivalent doses) within each operant AOP; and calculation of a mixture risk estimate (Figure 1).
**Step 1. Survey of existent toxicology data for mixture component chemicals**

The types of data available for individual chemicals that comprise a mixture often vary across a broad continuum, from those that have known adverse health outcomes with comprehensive annotation of associated signal transduction pathways, to those where virtually nothing is known beyond basic structure and limited physicochemical property data. Thus, the initial step in the AOP footprinting approach is the comprehensive and systematic collection and assemblage of all useful hazard and dose–response information (at all levels of biological organization spanning mechanistic to adverse health outcome[s]), toxicokinetic, and physicochemical property information, in this context are collectively considered “toxicology” information. Many approaches and resources for systematic collection, assembly and/or mapping of assessment relevant evidence are available in the public domain. For example, the EPA’s CompTox Chemicals Dashboard (https://comptox.epa.gov/dashboard/, accessed on 15 December 2022) provides diverse data, such as empirical or predicted physicochemical properties, existent hazard effect and dose–response (e.g., quantitative points-of-departure), and cell-based bioactivity (e.g., ToxCast) on as many as 1.2 M structures that could serve as a critical starting point for data assembly and evidence mapping across component chemicals identified in mixture.

Collection and collation of all available data in a standard reporting format that facilitates subsequent comparative evaluation among chemicals of a given adverse health outcome grouping, and, between chemicals of different adverse outcome groupings will provide the consistency and transparency needed for moving to the next steps below. More importantly this initial literature survey and evidence mapping step may help identify data gaps for chemicals and inform preliminary health outcome-based groupings.
**Step 2. Adverse health outcome-based grouping of mixture component chemicals**

The ab initio hazard/adverse health outcome data landscape of mixture chemicals will be a key first determinant in a grouping strategy where pre-existent hazard data will be the primary basis for initial assignment based on the weight-of-evidence (WOE). It should be noted that both qualitative and quantitative component chemical data are critical to the subsequent steps in the approach. Specifically, information that facilitates qualitative grouping or binning of component chemicals is clearly an important objective for subsequent AOP-based analysis. However, quantitative data that informs how chemicals behave with one another and/or with biological targets (across potentially multiple levels of biological organization) are paramount for key-event-based footprinting. For example, additivity or deviations from additivity among mixture chemicals may be evaluated at multiple points (i.e., Key Events [KEs] and [Key Event-Relationships [KERs]) along an AOP; as such, characterization of what magnitude of change in say KE1 impacts KE2 in an AOP is just as important as the qualitative observation that KE1 precedes KE2. Optimally, an AOP construct and the corresponding quantitatively annotated AOP (i.e., qAOP) would be identical however it has been observed that a qAOP model may not necessarily have the same pathway structure as the AOP on which it is based [24].

Importantly, mixture chemicals with more replete databases are likely candidates to serve as an ‘index chemical’ or ‘AOP anchor’ for a given AOP grouping (step 5 below); it should also be noted that mixture chemicals with WOE supporting association with multiple health outcomes warrant assignment to all relevant adverse health outcome groupings. Although it would be optimal to make health outcome grouping decisions for mixture component chemicals based on same/similar study designs and durations, indications of same/similar phenotypic effects or non-apical biological perturbations in common cell types, tissues, organs, and/or systems are key, irrespective of differences in in vivo and/or in vitro repeat-exposure study design; single dose or concentration/acute duration studies are of limited utility for grouping purposes as the toxicity of a mixture component chemical following long(er)-term repeated exposure is more informative for human health risk assessment purposes. Clearly expert judgment will be involved in grouping decisions based on study type and duration across component chemicals; transparent rationale is essential to support grouping decisions. For most mixtures, there will be some component chemicals that have sufficient apical effect data to support assignment to at least one adverse health outcome grouping. However, invariably there will be mixture component chemicals that lack repeat-exposure study data for proper grouping assignment at this step. For component chemicals with in vitro NAM data only (e.g., cell-based bioactivity assay data), decisions for health outcome group membership will likely be limited to those cell types in which the chemical was evaluated (e.g., bioactivity in hepatocytes in culture merits membership in a “liver” health outcome grouping). For those mixture component chemicals that are essentially devoid of any useful hazard/dose–response data, across any level of biological organization, it may be prudent to integrate other NAM platforms such as structure-activity/read-across to identify suitable analog chemical(s) for which hazard (i.e., adverse health effect domain) and dose–response data (e.g., point-of-departure; effect doses) might be adopted as surrogate for the data-poor target chemical [25]. Alternatively, algorithmic structure-activity tools and approaches such as Generalized Read-Across (GenRA) might be leveraged to make direct predictions of these hazard and dose–response parameters [26,27]. All mixture chemicals of concern, regardless of adverse health outcome data status, move to step 3.
**Step 3. Toxicokinetic profiling of mixture component chemicals**

Environmental mixture chemicals may range from low to high rates of absorption into systemic circulation, limited to broad tissue/organ distribution and bioavailability, and being virtually impervious to biotransformation in exposed human/mammalian populations up to generation of diverse arrays of primary, secondary, and tertiary metabolites. Characterization of the landscape of absorption, distribution, metabolism, and elimination (ADME) properties (i.e., toxicokinetics [TK]) of mixture chemicals can be highly informative for grouping purposes. For example, profiling the array of parent and metabolite species resultant from biotransformation can inform comparative TK evaluation between mixture chemicals. Chemicals with similar TK profiles, in particular similar metabolic patterns/profiles, warrant consideration for grouping even in the virtual absence of adverse health outcome data under step 2 above; or alternatively, TK data could be considered in conjunction with validated NAM data to better inform or refine adverse health outcome group membership for relatively data-poor mixture chemicals. Additionally, although it will not be discussed at length here, it should be noted that abiotic degradation of chemicals in environmental media is an important consideration that should not be ignored as this fate pathway can profoundly impact mixture component chemical speciation and/or proportion in potential exposure sources. Specifically, many environmental chemicals are altered in various environments (e.g., high/low oxygen; heat; UV light; pH, etc.). As such, it is prudent to ascertain, if possible, what mixture chemical species are represented in exposure sources (e.g., surface/ground water; soil); this information could be critical in subsequent characterization of toxicokinetics in human populations.

In this step, TK relevant information such as empirical or predicted absorption and elimination coefficients, distribution parameters (e.g., volume of distribution; blood or tissue half-life; etc.), plasma protein binding affinity, and in particular any information regarding metabolite profile (from studies assembled in step 1) are compared between those chemicals that were assigned to a hazard grouping in step 2 and those mixture chemicals that could not be assigned due to lack of available adverse health outcome data (or NAM data that inform potential health effects). The primary objective is to use the evaluation of similarities and differences in TK properties among mixture chemicals to inform potential assignment of ‘data-poor’ components to an adverse outcome grouping. That is, based upon expert judgment, any ‘data-poor’ mixture chemical with a TK profile that is similar to one or more candidate AOP anchor chemical(s) across assembled adverse outcome groupings could be assigned to a given grouping and the qualitative and quantitative toxicity data for the anchor chemical could be used to augment hazard WOE and/or dose–response may be adopted as surrogate for use in the subsequent steps of the approach. Any orphan mixture chemicals that remain unassigned to an adverse outcome grouping at the conclusion of this step are flagged for data need(s) and removed from further consideration in the AOP footprinting approach.
**Step 4. Assembly of Adverse Outcome Pathway(s) for a mixture of chemicals**

At this stage in the approach, there should be some general understanding of the landscape of adverse health outcomes potentially associated with exposure to a given mixture of chemicals. All mixture component chemicals with sufficient in vivo health outcome, NAM, and/or toxicokinetic information should have been assigned to one or more adverse outcome grouping(s) (in steps 2 or 3). In this step, mixture chemicals are associated at the level of AOP. Specifically, all candidate AOPs associated with a given adverse health outcome are reviewed and collected from extant sources such as the AOP wiki (https://aopwiki.org/, accessed on 2 December 2022). While it would be optimal if candidate AOPs are final or approved by an authoritative body such as the Organization for Economic Cooperation and Development (OECD), it is recognized that information underpinning pathway events may only occur in a draft or preliminarily proposed AOP construct. In some cases, there may not even be an AOP available that formally organizes biological data, identified for potential footprinting analysis in step 5. The user of the AOP footprinting approach will need to make decisions about inclusion or exclusion of ‘AOP(s)’ on a case-by-case basis; transparent communication of the evidentiary basis for AOP identification, and any associated uncertainties, will be key.

It should be noted that there may be some variations in effect description or vernacular in studies assembled (under step 1) versus the standard adverse outcome ontogeny employed in the AOP wiki. In most cases, the proper adverse outcome linkages should be apparent however expert judgment may be needed in edge cases (e.g., ‘hepatocellular injury’ versus liver ‘apoptosis’ or ‘necrosis’). In such instances, transparency in decisions regarding adverse outcome identification, as per AOP standard designation, would need to be clearly communicated in application of this approach. In the AOP wiki, specific health outcomes can be searched under the “view content” section and AOPs can be reviewed in tabular sequence and as part of proposed AOP network visuals. Each AOP identified as having involvement in a specific adverse outcome grouping, is subjected to AOP footprinting in step 5.
**Step 5. AOP footprinting**

Important considerations for any evaluation of human health pathway-based information include being clear about: (1) the beginning and end (i.e., termini) of a signaling pathway; and (2) directionality of the signal transduction. The scientific community does not have an accepted standard approach for identifying what constitutes beginning and end, upstream/downstream, “apical”, etc., as such, parameters need to be clearly communicated for application context. For the purposes of AOP footprinting, the beginning/starting point is considered that event that translates a dose or concentration into some subsequent biological perturbation or action in a target tissue. In AOP parlance this is commonly referred to as the molecular initiating event (MIE). An adverse outcome (AO) or effect is considered the end or phenotypic terminus of an AOP at the organism level. The movement of signal from an MIE through one or more KE(s) to an AO, in this approach, is considered moving “downstream”. Therefore, movement along the source-to-outcome continuum of a given AOP from MIE toward AO will be referred to as “forward” or “downstream” and conversely, moving from any node or event in a retrograde fashion (e.g., AO toward an immediately upstream KE) is considered to be moving “backward” or “upstream”. Under those parameters, the overarching principal of AOP footprinting is the stepwise profiling and comparison of AOPs between the group anchor chemical and all other candidate members of the AOP grouping at the level of key events *starting at the adverse health outcome and moving backward from the most downstream key event to the molecular initiating event*. The general steps of this AOP footprint approach are: (5.1) comparatively evaluate all existent AOPs suspected of involvement in a given adverse health outcome (identified in step 4 above) at the level of key events (Figure 2).

Beginning at the most downstream key event(s) for each health outcome specific AOP, systematically compare key events across to identify commonalities (i.e., biological perturbations that might be a convergence point for two or more AOPs); continue “up” (i.e., retrograde) each AOP toward the MIE(s) and identify each instance of common key event across AOPs. If at the conclusion of this comparative step there are no evident convergence points across AOPs (or within an AOP network), then each AOP will move to the next step as individual pathways; (5.2) for each AOP, identify anchor chemicals that have the most replete biological databases (ideally spanning from MIE to AO) where and when feasible; these will likely be the same mixture chemicals identified as candidate anchor chemicals based on robustness of adverse health outcome data in step 1; it should be noted that an AOP anchor chemical may not necessarily be the most potent among the group but rather the most well studied or characterized biologically as relevant for the target health outcome; (5.3) based on data-driven decisions made in 5.1, identify the most downstream key event(s) within a given AOP for an anchor chemical with weight-of-evidence (WOE) supporting a causal relationship to a given adverse outcome (i.e., the adverse health outcome either will not occur without said key event or at least the incidence and/or magnitude of the outcome is significantly diminished without it); *this key event is the ‘footprint’ for a given AOP*; (5.4) mixture chemicals within each adverse health outcome grouping are then evaluated for qualitative evidence of key event footprint(s) and assigned to the appropriate ‘footprint’ category based on similarity to the AOP anchor chemical; and (5.5) the AOP footprint dose- or concentration-response relationship(s) for each mixture chemical within a footprint category are then used to evaluate mixture dose additivity for that AOP via a component-based mixture assessment approach (e.g., relative potency factor; see step 6 below).
**Step 6. Component-based Mixtures Assessment: Relative Potency Factors**

The Relative Potency Factor (RPF) is one of several component-based mixtures assessment approaches applied under an assumption of dose additivity [8]. A key criterion for application of RPFs is that each mixture component chemical has dose–response data for the same toxicological effect or mode of action available for comparison to an index chemical (i.e., AOP group anchor). Additionally, application of the RPF method typically requires that the shape and slope of the dose–response functions for each mixture component are congruent. This is because scaling potency for an effect between two chemicals is treated as if mixture chemicals are dilutions of one another, at least within the range of effect or response of interest (i.e., doses at maximal effect or below No-Observed-Adverse-Effect-Levels [NOAELs] would not be informative). An important nuance that bears consideration is if a mixture or specific AOP grouping consists of partial receptor agonists. Specifically, for chemical mixtures with components that demonstrate affinity for receptor binding (e.g., Peroxisome Proliferator Activated Receptor (PPAR), Pregnane X Receptor (PXR), Liver X Receptor (LXR), Estrogen Receptor (ER), etc.), those mixture components with dose-effect levels that exceed the maximal effect of the least potent member present in the mixture, cannot be calculated. This is particularly challenging for mixtures containing partial receptor agonists/antagonists where chemicals often have differing maximal effects for a given receptor-based AOP. Scholze et al. [28] proposed a pragmatic solution that extrapolates the toxic units of partial agonists to effect levels beyond their maximal efficacy. This adjustment may be considered for mixtures that include components with data suggesting or demonstrating partial receptor agonism/antagonism.

Recognizing that quantitative exposure-response information assembled for members of a given AOP footprint grouping may derive from studies of different type (e.g., in vivo or in vitro) and/or duration (e.g., short-term/subacute, subchronic, or chronic), the first sub-step in step 6 of the approach is to harmonize the component chemical dose metric. Optimally, for chemicals in an AOP footprint grouping, all doses are converted to a human equivalent dose (HED). For example, the toxicity of some group members may have been evaluated using feed or water exposures (e.g., ad libitum drinking water) in experimental animals, commonly reported in ppm or mg/L, respectively. Study- or effect-based Lowest-Observed-Adverse-Effect-Levels (LOAELs) or benchmark doses (BMDs) (Benchmark Dose (BMD): A dose that produces a predetermined change in response rate of an effect (called the benchmark response or BMR) compared to background [29].) from traditional experimental animal assays would need to be converted to a corresponding HED using duration, species, age/weight, and sex-specific dosimetric adjustment factors [30,31]. Further, the target-site exposure may be different depending on what KE(s) are determined as critical in the footprinting analyses. Multi-scale dosimetry may be necessary to better contextualize the dose–response(s) across levels of biological organization. Clear communication of assumptions, parameters, and corresponding target dose outputs are paramount.

For AOP footprint membership based on in vitro bioactivity (e.g., cell-based assays) data, concentration-response information will need to be converted to corresponding estimated human in vivo exposure doses using reverse dosimetry and in vitro-to-in vivo extrapolation (IVIVE). IVIVE methods have been published broadly in the recent past [32,33,34]; also see https://ntp.niehs.nih.gov/whatwestudy/niceatm/comptox/ct-ivive/ivive.html, accessed on 5 December 2022), as such a detailed description of the approach is not provided here. In general, IVIVE incorporates high(er)-throughput in vitro toxicokinetics (e.g., protein binding; intrinsic hepatic clearance; unbound fraction in plasma), and in some applications, human population level biological variability, into models to predict in vivo external exposures [32]. The primary objective of reverse dosimetry applications of IVIVE is to convert bioactive concentrations from in vitro assays into approximately equivalent in vivo exposure doses (e.g., mg/kg-day). The initial dose conversion sub-steps mentioned here are imperative for harmonization of AOP footprint dosimetry for mixture component chemicals within a grouping for subsequent RPF calculations. For any mixture component chemical that does not have information available to support an IVIVE/reverse dosimetry conversion, a number of user-based options may be prudent such as using surrogate data from another member of the AOP grouping, especially those that are similar across structural, physicochemical, TK and/or biological similarity contexts. Alternatively, short-term experimental assays (e.g., “short-term” refers to in vivo exposures of greater than 24 h but less than 4 weeks in commonly used laboratory rodents; some literature refers to this duration as “subacute”) could be performed to generate kinetic data necessary for IVIVE/reverse dosimetry.

A related key consideration for RPF application in this approach is that comparison to an AOP footprint anchor chemical may be informed not so much by a point estimate, limited by experimental doses/concentrations tested (e.g., LO[A]EL for traditional study-based footprint KE vs. LO[A]EL for an in vitro assay-based footprint KE), but rather by a BMD. This is critical for RPF calculations as BMD modeling integrates the entirety of a given dose- or concentration-response dataset and would necessitate the identification of a dose associated with an AOP footprint event at an a priori determined benchmark response (BMR), for example, BMR50 (e.g., dose at an AC50, EC50, or IC50) (AC50 or EC50 = concentration at which a chemical induced half-maximal effect compared to positive control; IC50 is simply the converse; a concentration at which activity is reduced by 50% compared to positive control.) or biologically based BMR_X_ in the low(er)-dose region. In practice, it would be optimal to calculate and present NAM-based RPFs (RPF_NAM_) within an AOP footprint grouping based on two BMRs (i.e., RPFs based on BMDs at a data-driven/expert-determined BMR for the footprint event, and RPFs at a BMD based on a default such as BMR_50_ as a comparison) (Table 1).

Harmonizing the dose metric and utilizing BMDs optimizes comparison of the dosimetry across chemicals in an AOP footprint grouping irrespective of the data type or source (e.g., traditional assay- or NAM-based). In general, mixture components with dose–response functions that are significantly different from an AOP group anchor should not be included, as the interpretation of potency for effect/bioactivity will be inaccurate. However, since environmental chemical bioactivity profiles are typically diverse, a component chemical should then be considered within the context of other AOP footprint groupings to ascertain if alternative footprint dose–response data are more suitable for RPF application. Further, the approach does not preclude use of effect-based LOAELs, but BMDs are preferred to ensure that same response point along dose-response functions are used across AOP group members. It should also be noted that BMDs, that is, the central tendency estimates, are suggested rather than a lower statistical bound of the BMD (i.e., BMDL) since confidence intervals associated with some effects across different levels of biological organization can be quite large and may skew quantitative potency comparisons between AOP footprint member chemicals.

Once the dose- or concentration-response data have been converted/harmonized to human doses and BMD modeled for each member within an AOP footprint grouping (i.e., those group members with congruent dose–response shape/slope), RPF_NAM_(s) for each component chemical of a given grouping are calculated as follows in Equation (1):(1)$$RPF_{NAM}\;for\;n^{th}\;Mixture\;Chemical=\frac{BMDx\;for\;AOP\;footprint\;Anchor\;Chemical}{BMDx\;for\;n^{th}\;Chemical\;in\;AOP\;footprint\;group}$$

Since the units in the numerator and denominator of the ratio are the same (e.g., human equivalent BMDs), the resulting RPF_NAM_ is a unitless numeric. In this approach, the RPF_NAM_ represents the potency of the *n^th^* member chemical for activity associated with the AOP footprint relative to that of the AOP anchor chemical, for that AOP grouping. An RPF_NAM_ of 1 for example would indicate that a member chemical is equipotent to the anchor chemical for an AOP footprint event. An RPF_NAM_ of 0.5 would indicate that the *n^th^* member chemical is half as potent as the AOP anchor for the footprint event; conversely, an RPF_NAM_ of 2 would indicate that the *n^th^* member chemical is twice as potent as the AOP anchor.

The RPF_NAM_(s) for each AOP footprint group member are then used in conjunction with empirical or modeled exposure information, for each corresponding mixture component, in units consistent with the route (e.g., mg/kg-day for oral; mg/m^3^ for inhalation), to calculate a NAM-based AOP anchor chemical equivalent dose (ACED_NAM_) as presented in Equation (2):(2)$$ACED_{NAM}=RPF_{NAM}\times AOP\;footprint\;member\;exposure\;\left(dose\;or\;concentration\right)$$

The AOP-specific ACED_NAM_s for each footprint member are calculated and then summed within the grouping. This summed ∑ACED_NAM_ represents a total dose of the anchor chemical for a given AOP footprint. An AOP group-specific ∑ACED_NAM_ is then multiplied by the ratio of the BMD_X-HED_ for the apical health effect of the AOP anchor chemical to the BMD_NAM-HED_ (note: the user needs to be clear if a data-driven BMD or a default BMD_50_ is used) for the footprint event of the AOP anchor chemical as presented in Equation (3):(3)$$ACED=\sum^{\;}ACED_{NAM\;}\times \frac{Apical\;effect\;BMD_{X-HED\;}for\;AOP\;anchor\;chemical\;}{BMD_{NAM-HED}\;for\;AOP\;footprint\;for\;anchor\;chemical}\;\;$$

The resulting ACED represents the estimated AOP-specific contribution of mixture components to the overall risk of the specific health effect but represented as a dose scaled for potency, relative to the AOP anchor chemical, across different levels of biological organization (e.g., data-poor component chemical[s] in vitro to AOP anchor chemical in vivo). This process is applied across all operational AOP groupings for a specified health effect. The ACEDs across AOP footprint groupings are then summed to arrive at an ACED_MIX_ for the specific health effect. At this juncture, the user has two options: (1) Use the ACED_MIX_ in proper units for the purpose (e.g., mass/volume for a water concentration) to compare directly to a Health-Based Benchmark (e.g., oral Reference Dose [RfD], Health Advisory, or Maximum Contaminant Level Goal [MCLG]) for an AOP anchor chemical, which presumably would be the most health protective value among anchor chemicals for the specified apical health effect. That is, the ACED_MIX_ for a given health effect should be the same effect (or at least similar; for example, a specified developmental effect that is part of a constellation or syndrome of developmental effects) that underpins the derivation of a corresponding health-based benchmark value. The approach may however be applied across multiple health effect domains for comparison purposes where anchor chemical health-based benchmark value(s) and AOP group data are available. An ACED_MIX_ that exceeds a corresponding anchor-chemical specific health-based benchmark would suggest potential for adverse health outcome(s) associated with exposure to the mixture;
*ACED_MIX_ is <, =, or > an existent media- or site-specific Health-Based Benchmark*(4)
or, (2) Rather than sum ACEDs to arrive at an ACED_MIX_, use each AOP anchor chemical ACED to map to the corresponding specific apical health effect dose–response curve to estimate the response associated with the mixture chemicals in the specific footprint grouping (Figure 3). Then, based on an assumption of independence (i.e., response addition) across operational AOPs, individual AOP-specific responses are summed to arrive at an estimated mixture-level response for a specified health effect.

As with any chemical evaluation or assessment, discussion of uncertainties involved in an AOP footprint analysis are requisite. For clarity, it is suggested that distinctions are made between challenges associated with the qualitative aspects and quantitative uncertainties that may impact interpretation in predicted joint toxicity of a mixture. Generic examples of the former might include AOP membership, pathway-based WOE, absence of an established AOP consistent with anticipated biology of the AO/health effect, lack of AOP anchor chemical(s), etc. Examples of quantitative uncertainties might include lack of data to inform multi-scale dosimetry, unanticipated dose–response influence of mixture component-chemical KEs across a network of AOPs (e.g., presence of chemical in one AOP impacts the kinetics of a chemical in another AOP), dose-dependent transitions in biology, etc. Some standardized reporting format and pathway-based ontogeny would benefit clear communication of the various uncertainties potentially included in an AOP footprinting analysis.

## 3. Example Application of AOP Footprinting to a Hypothetical Chemical Mixture

To illustrate how AOP footprinting may significantly advance an approach for chemical mixtures risk assessment, consider a hypothetical mixture of six chemicals (A–F). Under steps 1–2 of the AOP footprint process, chemicals A and D were identified to be active in the liver based on available in vivo apical effect level weight-of-evidence (WOE) for significantly increased incidence of hepatocellular injury/death across species and exposure durations for the oral route. As such, chemicals A and D are identified as candidate AOP group anchors (Figure 4). Chemicals B, C, E, and F were found to lack any traditional in vivo assay-based information for adverse health outcomes in the liver. Steps 1–3 of the approach however revealed NAM (e.g., cell-based bioactivity) data available for these specific mixture chemicals that suggest the liver as a potential target of toxicity.

Following evaluation of NAM data across all mixture chemicals in step 4, three potential operant liver AOPs were identified (e.g., AOPs 273 [Mitochondrial complex inhibition leading to liver injury], 278 [IKK complex inhibition leading to liver injury], and 27 [Cholestatic liver injury induced by inhibition of bile salt export pump]; see AOP Wiki at https://aopwiki.org/, accessed on 5 December 2022) and subsequent AOP footprinting in step 5 qualitatively supported the clustering of mixture components A, B, and C (Liver AOP 278), and D and E (Liver AOP 273) as logical footprint groupings. Upon closer examination of the available NAM data, chemical F appeared to be inducing an increase in hepatocellular injury via an AOP (i.e., Liver AOP 27) that is distinctly different from footprint group chemicals A–E (Figure 4). While chemical F may indeed be contributing to an overall mixture risk of an adverse liver outcome, the AOP footprint approach may not be suitable for chemical F since there were no apical health outcome data for this mixture component, or an appropriate AOP anchor chemical. Chemical F would need to be flagged for further data need(s). Short(er)-term repeat-dose animal studies may be optimal as they are (relatively) animal sparing and expedient and may provide information that facilitates identification of chemical F as a potential anchor chemical. In the event that sufficient data were available for chemical F, contribution of AOP 27 to the overall interpretation of mixture risk of liver injury might be accomplished by calculating the ACED associated with chemical F and AO 357, and through an assumption of response addition (i.e., independent action), integrating with the ACEDs for AOPs 278 and 273 in vitro.

For the Liver AOP 278 and AOP 273 footprint groupings, chemicals A and D are confirmed as the anchor chemicals, respectively, based on robustness of WOE for AOP events in step 5 (as well as elicitation of the specified adverse liver health outcome; identified under steps 1–2). Systematic AOP footprinting for anchor chemical A and the proposed group members B and C, for AOP 278, revealed a common AOP ‘footprint’; specifically, in the hypothetical mixture, the liver injury footprint event for mixture chemicals A, B and C is *increased hepatocellular injury/death* as it is the most downstream key event that is shared by all three components and for which dose- or concentration-response data are available. For liver AOP 273, although the molecular initiating event (MIE) and early KEs are distinctly different from those of AOP 278 (see Figure 4), the footprint event for chemicals D and E was also identified as *increased hepatocellular injury/death.*

Under step 6, relevant dose- and/or concentration-response data for AOP footprint members of each grouping are converted to human equivalent doses. Obviously if human epidemiological data had been available, the doses could have been used directly. However, since only traditional experimental animal assay data are available for AOP anchor chemicals A and D, dosimetric adjustment to a human equivalent dose would proceed based on current U.S. EPA risk assessment practice. For example, in the Recommended Use of Body Weight^3/4^ as the Default Method in Derivation of the Oral Reference Dose [30], the U.S. EPA endorses a hierarchy of approaches to derive human equivalent oral exposures (i.e., human equivalent doses; HEDs) using data from laboratory animal species, with the preferred approach being physiologically based toxicokinetic modeling. Other approaches might include using chemical-specific information, without a complete physiologically based toxicokinetic model. In the absence of chemical-specific models or data to inform the derivation of human equivalent doses, the U.S. EPA endorses BW^3/4^ as a default to extrapolate toxicologically equivalent doses of orally administered agents from laboratory animals to humans.

For NAM data such as in vitro cell-based bioactivity, cell culture concentrations can be converted to equivalent human external exposure doses using in vitro to in vivo extrapolation (IVIVE) and reverse toxicokinetics (rTK). The resulting human administered equivalent dose–responses (AEDs for NAM data; HEDs for experimental animal assay data) are subsequently BMD modeled, where/when amenable to modeling, at a data-driven/expert selected BMR (e.g., BMR_10HED_) for change from control for endpoints such as incidence in liver injury, or BMD_1SDAED_ for NAM data such as cell bioactivity, as well as BMR_50AED_s for NAM data (Table 2).

RPF_NAM_s, based on the ratio between in vitro cell bioactivity data (e.g., hepatocellular death; KE 55 in Figure 4) for the AOP footprint event for the AOP anchor chemical and another member of the AOP grouping, are then calculated within each AOP footprint grouping, as per Equation 1, using data-driven BMD_1SDAED_s (Table 3). The RPFs are then multiplied by the empirically determined or measured (or alternatively, predicted) exposure for each AOP footprint member to obtain a corresponding ACED_NAM_ for Liver AOP 278 and AOP 273 (Table 3). The footprint group ACED for Liver AOP 278 is then derived by multiplying the sum of the ACED_NAM_s by the ratio of the BMD_10HED_ for increased incidence of liver injury (i.e., AO 1549) associated with anchor chemical A to the BMD_1SDAED_ for increased hepatocellular death, also for anchor chemical A (Table 3). The same calculation is performed for Liver AOP 273 however the BMDs used are based on anchor chemical D. The anchor chemical equivalent dose for the mixture (ACED_MIX_) equals the sum of the ACEDs for Liver AOP 278 and AOP 273; in the hypothetical example ACED_MIX_ = 0.56 + 0.4 = 0.96, rounded to one significant digit of 1.0 mg/kg-day.

For illustrative purposes, mixture chemical A has an existent oral reference dose (RfD in mg/kg-day) of 7.0 × 10^−4^ based on increased incidence of liver injury (observed in subchronic duration studies in rats) derived from a point-of-departure (POD) of 0.2 mg/kg-day and composite uncertainty factor (UF) of 300 (comprised of uncertainties associated with human population variability [UF_H_ of 10]; extrapolation of toxicity from animal-to-human [UF_A_ of 3]; and subchronic-to-chronic duration extrapolation [UF_S_ of 10]). The RfD represents a daily oral dose over a lifetime that is anticipated to result in no adverse effect on the liver; doses above the RfD are anticipated to result in adverse effect(s), particularly in the liver. The ACED_MIX_ of 1.0 mg/kg-day for increased liver injury can then be compared to the RfD (7.0 × 10^−4^ mg/kg-day) for chemical A to ascertain if there is or is not concern for liver injury due to exposure to the mixture of chemicals A–E. Clearly in the hypothetical example, the total mixture ACED_MIX_ exceeds not only the RfD but even the POD for liver injury, suggesting significant concern for liver effects associated with exposure to the mixture. The potential alternative approach of mapping the ACEDs for components A and D to their respective dose–response functions for liver injury, under and assumption of pathway independence (i.e., response addition), would not be advisable as there is a clear pathway convergence of AOPs 278 and 273 at the AOP footprint event (i.e., KE 55; see Figure 4). Had AOP 27 been integrated into the analysis, there may have been an opportunity to employ this alternative approach as this specific pathway does not share KEs with AOPs 278 or 273 but likely contributes to the ultimate liver AO.

## 4. Discussion

Human biomonitoring studies for decades have indicated that we as individuals and members of populations are exposed to mixtures of chemicals via the food or water we ingest, the air we breathe, and/or human activity that puts us into contact with other environmental matrixes or exposure sources (e.g., occupational/non-occupational dermal; etc.) [35,36,37]. The CDC’s NHANES database has for years confirmed the presence of mixtures of chemicals in human blood samples (please refer to https://www.cdc.gov/nchs/nhanes/index.htm, accessed on 11 December 2022). Therefore, assessment of the potential joint toxicity of chemicals co-occurring in humans is of paramount importance. Additionally, other than well-characterized chemical classes such as dioxins/furans and PCBs/PAHs, formal EPA assessment of mixtures of chemicals has been virtually non-existent to date. The primary limiting factors are: (1) Lack of hazard and dose–response data for whole mixtures of concern; and (2) The reliance of component-based mixtures assessment methods on traditional human epidemiological or experimental animal apical effect data. This highlights the necessity for leveraging NAMs in mixtures assessment approaches.

To date, integration of NAMs into risk assessment has predominately been in the form of qualitative data-gap filling or augmenting hazard-based WOE. More recently, research studies have focused on quantitative applications of NAMs such as identification of cell-based and short-term whole animal bioactivity or -omics based points-of-departure for potential use in derivation of cancer or non-cancer toxicity values, or risk-based metrics such as a bioactivity-exposure ratios [38,39]. This type of work demonstrates opportunities to significantly advance the science of hazard and dose–response assessment, even in the virtual absence of traditional data types. In a mixtures assessment context, the integration of qualitative and quantitative data across different levels of biological organization, including NAMs, under a structured pathway annotation construct such as Adverse Outcome Pathway (AOP) may provide a foundation for fundamentally shifting the chemical mixtures risk assessment paradigm [40].

The proposed advancement, suggested in general by this AOP footprinting approach, in using different data streams/types in chemical mixtures risk assessment is three-fold; for one, leveraging NAM data places less reliance on the availability of traditional long(er)-term animal bioassay data which is more resource intensive and less timely. As such, there are greater opportunities to incorporate more chemicals into a mixture evaluation by using NAM (e.g., in vitro and/or short-term in vivo bioactivity) data. Second, existent component-based mixtures methodology consistent with U.S. EPA guidance and practice is suited to evaluate joint toxicity of chemicals using new(er) data types; no new mixtures dose–response approaches are needed. That is, approaches such as the relative potency factor are ideal for evaluation of mixture component chemical dose–response no matter where along an AOP continuum the potency comparisons are made, assuming some level of congruency of dose–response shape and slope across chemicals. For those mixture component chemicals with statistically different dose–response functions, an alternative method may need to be considered based on assumptions of additivity. Lastly, in contrast to current AOP theory which posits a stressor agnostic description of a pathway from the MIE to AO, the footprinting approach presented in this work is optimized by identification of well-characterized (hazard and dose–response) chemicals as the “anchor” for each causal AOP. In essence, the AOP footprinting method integrates elements of AOP and “mode of action” where identification or description of the source to health outcome continuum is consistent with both constructs; it is only in the systematic identification of and dependence on an AOP anchor chemical that is more MOA-like. However, this is necessary so that key event data for members of a given AOP footprint grouping can be contextualized with respect to the anchor chemical such that estimation of an apical health outcome for the mixture as a whole is possible.

A critically important aspect of the AOP footprinting approach is the bottom-up evaluation of an AOP. Specifically, whether it be for evaluation of individual chemicals or for mixtures assessment, it is posited that a more logical approach to assessment application(s) is to base qualitative hazard and quantitative dose–response interpretations on key events that are more proximally located to the actual health outcome of concern, regardless of level of biological organization from which data are sourced. While MIEs and earlier key events are important to characterize, particularly if that is the only bioactivity data available for a data-poor chemical, the qualitative diversity of key events earlier in an AOP may often suggest biological profiles that are different from chemical to chemical. That is, for many signal transduction pathways the “signal” (e.g., MIEs or early key events) may appear to be dissimilar among chemicals postulated to impact the same health outcome, however nearer to the terminus of the same signaling pathway(s) there is greater likelihood of observing similarity or likeness across chemicals due to pathway convergence. These convergence points are commonly more proximal to the effected phenotypic cell population, tissue, and/or organ associated with a chemical insult. As such, particularly when evaluating whether mixture chemicals belong to a similarity grouping (e.g., AOP footprint grouping), starting at the AO, or most downstream key event for which data are available, and working backwards “up” the AOP, for a given grouping of chemicals, should more often than not lead to identification of one or more key events (e.g., convergence points) as candidate AOP footprints on which joint toxicity of the mixture can be evaluated.

There will be challenges associated with practical application of the AOP footprinting approach such as general availability of data for mixture component chemicals including NAM-based data from validated methods, existence of approved AOPs to incorporate into the evaluation or flexibility to propose de novo AOPs, dose–response complexities (e.g., dose-dependent transitions in key event behavior; different shapes and/or slopes of dose–response functions between key events in an AOP, or between key events and the AO; etc.), amenability of dose–response data to benchmark dose modeling, and variable interpretations of dose additivity, or deviations from additivity (e.g., synergism, antagonism), along a given AOP. These issues are not unique to AOP footprinting or even the use of NAMs in general; the challenges will likely be case-specific and simply require transparent communication of assumptions and uncertainties in the analysis. Discovery of strengths and limitations of the AOP footprinting approach will only be realized through practical example applications, which will be the focus of work moving forward.

## Figures and Tables

**Figure 1 toxics-11-00037-f001:**
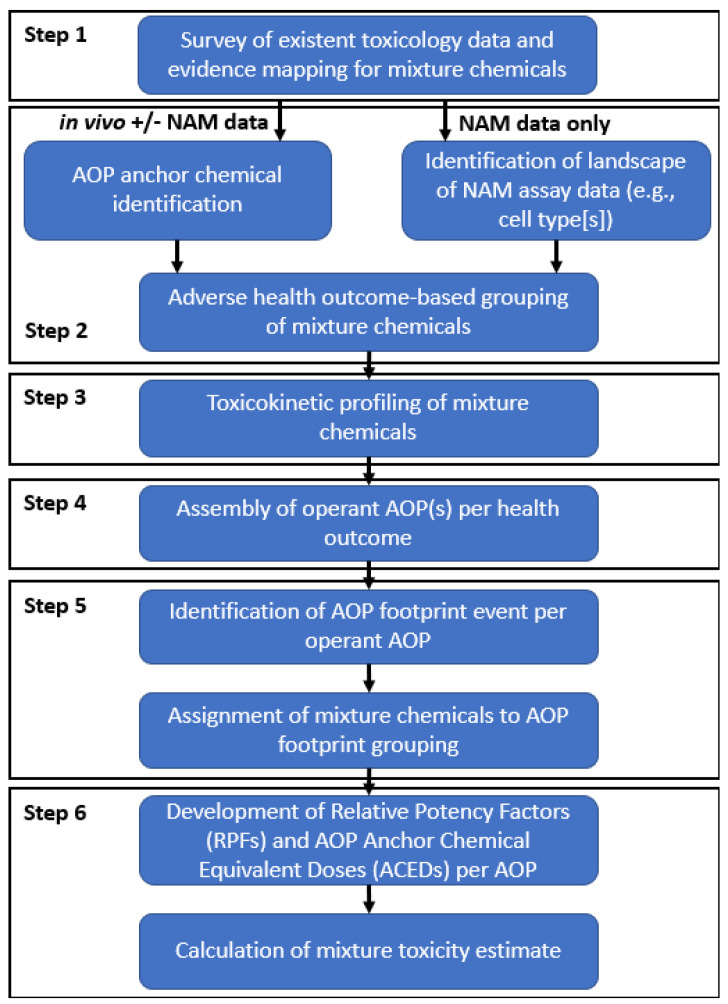
**AOP footprinting workflow.** The general workflow starts with the assembly of all available in vivo toxicity and in vitro NAM (e.g., cell-bioactivity; toxicogenomic) data and culminates in the calculation of a mixture risk estimate via the development and application of relative potency factors within each operant AOP identified for a given health outcome.

**Figure 2 toxics-11-00037-f002:**
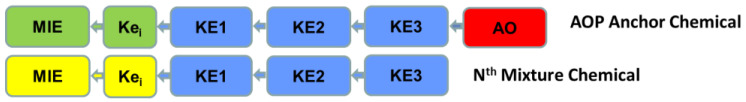
**General Adverse Outcome Pathway (AOP) footprint process.** For each individual AOP identified in step 4, in step 5, ‘footprinting’ entails the systematic retrograde evaluation of key events between an AOP anchor chemical and other members of the specific AOP grouping. The most downstream key event (i.e., most proximal to the adverse outcome [AO]) that is shared between the AOP anchor chemical and group members and has available dose-/concentration-response data, is identified as a candidate AOP ‘footprint’. It should be noted that although the hypothetical AOP shown is tabular/linear in sequence, for simplicity of interpretation, in practice there will likely be diversity in KEs the further upstream (i.e., toward earlier KEs and the MIEs) one moves up a given AOP.

**Figure 3 toxics-11-00037-f003:**
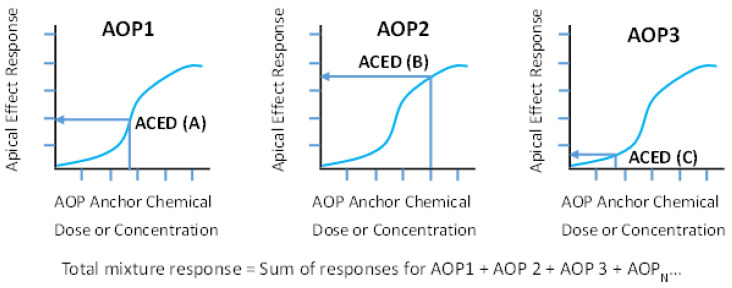
**Hypothetical apical effect dose–response curves for AOP anchor chemicals A–C.** Under step 6 of the AOP footprinting approach, under an assumption of independence (i.e., response addition), apical effect responses associated with each individual anchor chemical’s ACED across operant AOPs may be summed to estimate the total response associated with exposure to the mixture.

**Figure 4 toxics-11-00037-f004:**
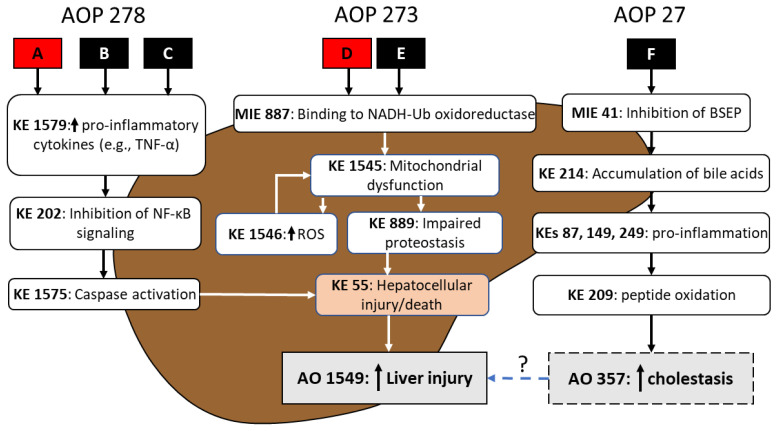
**AOP Network for hypothetical example mixture of components A–F.** Under step 4 of the AOP footprint process, based on traditional assay and/or NAM (e.g., evidence of bioactivity consistent with key events) WOE across the mixture components, three operant liver AOPs (278, 273, and 27) were identified for the specified health effect (e.g., increased liver injury). Data were also used to confirm selection of AOP anchor chemicals (e.g., mixture components A and D for liver AOPs 278 or 273, respectively) and to support assignment of data-poor components (i.e., chemicals B, C, E, and F) to the appropriate AOP grouping. Under step 5 of the process, an AOP footprint event is identified; in this hypothetical mixture example the liver AOP footprint is ‘KE 55: hepatocellular injury/death’ for both AOPs 278 and 273. Mixture component F is also a potential contributor to liver injury associated with exposure to the mixture however due to lack of AO data and identification of an AOP anchor chemical this component does not move on to step 6.

**Table 1 toxics-11-00037-t001:** **Suggested array of Benchmark Doses (BMDs) to calculate for chemical mixture members within an AOP grouping.** For a given AOP proposed to be involved in the elicitation of a specific adverse outcome, the human equivalent dose apical effect BMD(s) and NAM-based human equivalent dose BMDs should be calculated for all members of an AOP footprint grouping, where and when data are available. It is recommended to calculate BMDs based on both data-driven/expert-determined benchmark responses (BMR) appropriate for the type of data/effect, and a default such as a BMR_50_ as a comparison. ND = not determined.

AOP 1 Group Member	Apical Effect BMD_X-HED_	Data-Driven AOP Footprint-Based BMD_X-NAM(HED)_	Default AOP Footprint-Based BMD_NAM50(HED)_
**A**	BMD_X-HED_	BMD_X-NAM(HED)_	BMD_NAM50(HED)_
B	ND	BMD_X-NAM(HED)_	BMD_NAM50(HED)_
C	ND	BMD_X-NAM(HED)_	BMD_NAM50(HED)_

**Table 2 toxics-11-00037-t002:** **Component chemical BMDs for a hypothetical mixture of components A–E.** Human equivalent BMDs are modeled at a BMR appropriate for the apical effect type (e.g., BMR_10_ for incidence of liver injury). BMDs are calculated for the AOP footprint event(s) based on the converted administered equivalent dose (AED)-response (note: an AED is an estimated oral exposure dose that results in an internal steady-state concentration in humans consistent with the in vitro concentration associated with a biological perturbation or activity). The preferred NAM-based BMR is data-driven/expert-selected, in this example set at one standard deviation (1SD) change over corresponding control; the BMR_50_ is also provided as a comparator however it is not required to carry the default-based BMDs further unless desired by the user. The numerical values provided are hypothetical and for illustrative purposes only.

**AOP 278 Group** **Member**	**Apical Effect** **BMD_10HED_**	**Data-Driven AOP Footprint-Based BMD_1SDAED_**	**Default AOP Footprint-Based BMR_50AED_**
**A**	0.3 mg/kg-day	0.08 mg/kg-day	0.4 mg/kg-day
B	ND	0.01 mg/kg-day	0.08 mg/kg-day
C	ND	0.7 mg/kg-day	1.5 mg/kg-day
**AOP 273 Group** **Member**	**Apical Effect BMD_10HED_**	**Data-Driven AOP Footprint-Based BMD_1SDAED_**	**Default AOP Footprint-Based BMR5_0AED_**
**D**	0.02 mg/kg-day	0.03 mg/kg-day	0.08 mg/kg-day
E	ND	0.009 mg/kg-day	0.05 mg/kg-day

**Table 3 toxics-11-00037-t003:** **Calculation of RPFs, ACED_NAM_s, and AOP footprint-specific ACEDs for a hypothetical mixture of components A–E.** RPF_NAM_ = Data-driven anchor chemical BMD_1SDAED_ for AOP footprint/data-driven BMD_1SDAED_ for components in AOP footprint group; ACED_NAM_ = RPF_NAM_ × AOP footprint member exposure; ∑ACED_NAM_ = sum of group member ACED_NAM_s; AOP footprint group ACED = ∑ACED_NAM_ × apical effect (e.g., liver injury) BMD_10HED_ for the AOP anchor chemical/data-driven anchor chemical BMD_1SDAED_ for AOP footprint.

**AOP 278 Group Member**	**RPF_NAM_**	**Exposure** **(mg/kg-day)**	**ACED_NAM_** **(mg/kg-day)**	**AOP 278 Footprint Group ACED** **(mg/kg-day)**
**A**	1	0.09	0.09	
B	8	0.007	0.06	
C	0.1	0.02	0.002	
∑ACED_NAM_ = 0.15				**0.56**
**AOP 273 Group** **Member**	**RPF_NAM_**	**Exposure** **(mg/kg-day)**	**ACED_NAM_** **(mg/kg-day)**	**AOP 273 Footprint Group ACED** **(mg/kg-day)**
**D**	1	0.4	0.4	
E	3	0.05	0.2	
∑ACED_NAM_ = 0.6				**0.4**

## Data Availability

No data were derived de novo for this manuscript. All references to data sources proposed for use in the methodology have been cited or linked in-text.

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
