# Peer review of "Adverse Outcome Pathway ‘Footprinting’: A Novel Approach to the Integration of 21st Century Toxicology Information into Chemical Mixtures Risk Assessment"

_toxics, 2022, doi:10.3390/toxics11010037_

Round 1
Reviewer 1 Report
I must first sincerely apologise for the long delay in reviewing this important manuscript. However, I was not disappointed by its content. The manuscript is very well-written and demonstrates a thorough and deep understanding of the field. It presents a novel theoretical approach to integrating toxicological data derived from NAMs into an AOP framework for the purpose of mixture risk assessment. I only have a few minor suggestions for revisions before this can be accepted for publication.
- In the introduction, line 113, I would context the assertion that historically additivity has been observed for apical endpoints. The recent systematic review by Martin et al (2021) (https://doi.org/10.1016/j.envint.2020.106206) found that most mixture experiments assessing adherence to additivity hypotheses (dose-addition or independent action) have been conducted in vitro.
- In step 5, lines 325-348, the use of singular or plural when relating to key event footprint(s) or footprint could be disambiguated or explained further. Figure 1 depicts a linear AOP whereas previous and subsequent sections adequately explain that AOPs are not necessarily linear and could consist of a network of AOPs. How this would be implemented in the proposed approach could be expanded upon before the reader reaches the theoretical example. Upon reading this section, this was not entirely clear.
- The reliance on the relative potency factor (RPF) method in step 6 is well-noted and the author rightfully states that this requires that the shape and the slope of mixture components are similar (with anchor chemical). This in turn leads to exclusion of mixture components that do not fulfill this assumption. This is appropriately discussed as a limitation in the discussion section. Besides the potential for underestimating mixture effects by excluding components, I was confused by the fact that this grouping step seem to occur before doses of chemicals in an AOP footprint grouping are converted to Human Equivalent Doses. This may deserve further explanation. The work of Scholze et al (2014) (https://doi.org/10.1371/journal.pone.0088808) may also be of interest in this context.
- With reference to step 6.2 based on the assumption of independent action (l488-492), the author states on lines 671-672, that this case scenario could have been explored if AOP27 had been integrated into the analysis as this pathway does not share KE with AOPs 278 or 273. I am wondering about the evidence base for such a statement for an endpoint such as organ toxicity (as opposed to algal toxicity where the application of independent action has been demonstrated), i.e. that chemicals in a mixture acting via pathways that do not share a KE would adhere to independent action as opposed to concentration addition. If you are aware of such evidence, it would be important to cite it here.
- Do check that acronyms are defined on first apparition. I detected issues with GPCRs (l128), ND in tables, RfD on first apparition (l474), 1SD table 2.
Finally, as correctly stated in the conclusion, this is a theoretical approach and discovery of the strength and limitations of this interesting novel method will only be realised through its practical implementation. I hope to read your future work on this very topic in the near future.
Author Response
Thank you for the feedback on this manuscript. Please see attached memo.

Reviewer 2 Report
This is an important paper. It needs to be edited to provide readability.

Author Response

(The authors gave the same response as above.)

Reviewer 3 Report
Review of ‘Adverse Outcome Pathway ‘Footprinting’: A Novel Approach to the Integration of 21st Century Toxicology Information into Chemical Mixtures Risk Assessment
The paper is providing a general concept for integration of New Approach Methodologies (NAMs) and data created from NAMs into applications such as human health risk assessment following exposure to environmental mixtures. NAMs area of research is a growing field and there have been great improvements towards use of NAMs for human health risk assessment approaches instead of use of animal studies. Overall, this review is of importance to the field and has significant scientific importance. However, there are areas need more clarity to help reader t understand the overall approach and what ‘mixtures’ this approach is suitable for. Please see below for list of suggestions and comments to help improve the clarity in certain areas highlighted in the manuscript.
Abstract:
Line 13, there is a debate about how the validation is defined. Does the author mean analytical method validation for NAMs? It may be good to add couple of line about the definition of ‘validation’ in the intro section.
Line 14, it is not clear if the sentence (leveraging extant hazard and dose response…) refers to individual chemicals vs mixtures
Line 16, similar to above comment it is not clear if ‘advance risk assessment of chemicals’ refers to chemicals or chemical mixtures
Intro:
Line 29, ‘lack of dose response information’: l believe Earl Gray have studies on mixtures dose response. in addition, other papers look at the mixtures e.g., Dose Addition Models Based on Biologically Relevant Reductions in Fetal Testosterone Accurately Predict Postnatal Reproductive Tract Alterations by a Phthalate Mixture in Rats by Howdeshell et. al. I agree with the author there is limited data available to represent the 'real life' mixtures however, I would caution to use 'limited' instead of 'lack of'.
Line 29, ‘whole mixture’, it would be good to define 'whole mixture' vs 'individual component chemicals'. For mixtures world these are used constantly but someone who is new to the mixtures area of research, this may not be very obvious. There are publications refers to whole mixtures as well.
Line 32, ‘’real life’ mixtures’, it may be good to provide examples for these types of exposures.
Line 38, ‘chemical may transit...’, metabolism at the target site is also an issue since the chemicals may not stay as parent but metabolize to another product
Line 61, does the author consider botanicals (e.g., green tea, botanical supplements such as black cohosh etc.) as complex mixtures? or this specifically meant for environmental exposures only (e.g., woodsmoke exposures, drinking water contaminants etc.)
Line 61, does ‘environmental chemical’ refer to chemical or chemical mixtures?
Line 93, ‘NAMs platforms’, as a side comment, I have rarely seen genetox assays incorporated part of NAMs approaches. Are there any NAM platforms that utilizes genetox assays the author came across that maybe beneficial part of AOPs?
Line 94, ‘mixture exposure’, I think it would be beneficial to explain if the mixture exposures include botanicals. It is an ongoing effort and challenge to regulate botanical supplements that are considered complex mixtures and have bioactive known and unknown constituents. bioassay directed fractionation were commonly used in some of the alternative testing methods
Line 104-105, ‘may be dose or response additive…’, dose response additive? If later (dose or response) please give examples of what kind of response.
Line 112, ‘essentially independent’, does that mean they are independent (independent MOA and AOP) but still response is heightened dose proportionally?
Line 127-131, This article seems to be focusing on known complex mixtures rather than unknown complex mixtures (such as botanicals). A disclaimer at the beginning would be better to have the reader focus on more known environmental mixtures. There is a mention of environmental mixtures/exposures at the beginning but adding one or two additional sentences on the focus of mixture 'types' would be beneficial.
Line 151, ‘concentration-response…’, one key point could be external and internal exposure concentrations due to metabolism. Maybe earlier when ADME and TK properties are discussed, internal vs external exposure concentrations should be discussed. Since metabolism is a big factor when comparing AOP and target tissues, using metabolically active alternative methods vs non-actives were also challenging to evaluate the outcomes of the alternative approaches.
Method section:
General comment, it would be nice to have a graphic illustration of the method for AOP Footprinting to show each step. That will help reader to understand if the approach follows e.g., a waterfall diagram or something else. if each step is dependent to or independent from each other (e.g., end results of one step initiate or feeds into other step(s)).
Line 159, ‘…beyond and basic structure…’, this comes back to unknown components of environmental mixtures. can the author mention anything about the ones with largely unknown components/ Is there another approach for complex mixtures with unknown components?
Line 168, ‘…across multiples of chemicals’, not clear if individual chemicals or not.
Line 202, ‘…steady state…’, It is not clear if the author is saying steady state can be only reached following repeat exposure or limited to aid the interpretations of exposures especially when it comes to mixtures and human health. At times depending on exposure dose, accumulation and rate of clearance, the concentration of chemical(s) can reach to a steady state after single dose. granted repeat exposures provides better understanding about the toxicological outcomes and target organs, therefore it is critical to evaluate reversible vs irreversible damage caused by exposure. It would be good to explain the single vs repeat dose comparison when it comes to human health risk assessment purposes. NAMs also could be limited when evaluating long term repeat dose.
Line 220,’…regardless of adverse…’, what about the ones with unknown components but still suspected to have adverse health outcomes. Is that then a non-targeted analysis is done to try to figure out components
Line 221, is ‘the next step’ referring to step 3?
Line 224, I would recommend adding bioavailability to the list.
Line 233, ‘…pattern/species…’, Not clear if this means similarity across species? or within species e.g., male and female rat (no sex differences)
Line 263, title, ‘mixture of chemicals’, is this step based on known components of the mixture?
Line 310-314, step 5, bolded lines, if this is a reference it is not clear. Not sure if I understand the purpose of italicizing this. Is this a statement taken from elsewhere? If so please provide the reference.
Line 325, This section would be easier to follow if a different diagram (a decision tree) was chosen to walk the reader through. depending on the answers (e.g., evidence on convergence points across (or within) AOP, if the answer is 'yes' what to do and another option for the answer 'No' etc.)
Line 371, ‘water exposures’, drinking water exposure or oral gavage using water as a vehicle or is this referring to environmental exposures due to drinking water?
Line 375, Reference for HED, I would also recommend adding another reference for HED calculations, Reagan-Shaw S., Nihal M., Ahmad N. (2008) Dose translation from animal to human studies revisited. FASEB Journal: Official Publication of the Federation of American Societies for Experimental Biology, 22, 659–661.
Line 383, parentheses or reference bracket?
Line 386, ‘high(er)-throughput’, I would recommend using medium to high throughput
Line 388, ‘external exposures’, why external? e.g., plasma levels are internal dose.
Line 390, ‘in vivo doses’, this is a little confusing, does that mean internal dose or external exposure concentrations?
Line 390, recommend using mg/kg/day instead of mg/kg-day
Line 397, ‘experimental assays’, could the author define ‘short term experimental assays'. In vitro vs in vivo. If also the author can provide their input on modeling approaches instead of assay, what they would recommend? There are many models available there (some free some not) such as ADMET.
Line 407, BMRx, not sure if x referring to AC50 etc. or just a dose range?
Line 429-433, additional gap between sections
Line 433, I think it is important to define what dose and concentration mean. It gets confusing if one refers to external dose and other refers to internal dose. Same for human doses. Does the author refer to internal dose?
Line 501, please check the additional line between 501 and 503
Line 519, Additional gap between the 2 sentences
Line 530, Figure 3, I really like the example that walks through the steps. I think one clarification a reader might need is that if there is no liver AOP but there are other target organs for some of the compounds. How the author would go about it. e.g., avoid the ones that have different target organs and pathways to evaluate multiple AOP Networks? And when there is no information whatsoever about some of the chemicals, what is the next step. Eliminate those chemicals from the equation or use modeling approaches to make assumptions. Additional guidance would be beneficial to aid unknown components of the chemical mixtures.
Line 587, how informative is the human external exposure concentration when in fact what matters is the internal dose.
Author Response

(The authors gave the same response as above.)
